# Encapsulated *Bdellovibrio* Powder as a Potential Bio-Disinfectant against Whiteleg Shrimp-Pathogenic Vibrios

**DOI:** 10.3390/microorganisms7080244

**Published:** 2019-08-07

**Authors:** Haipeng Cao, Huicong Wang, Jingjing Yu, Jian An, Jun Chen

**Affiliations:** 1National Pathogen Collection Center for Aquatic Animals, Shanghai Collaborative Innovation for Aquatic Animal Genetics and Breeding, Shanghai Engineering Research Center of Aquaculture, Shanghai Ocean University, Shanghai 201306, China; 2Department of Animal Husbandry and Veterinary Medicine, Jiangsu Vocational College of Agriculture and Forestry, Jurong 212400, Jiangsu, China; 3Lianyungang Marine and Fisheries Development Promotion Center, Lianyungang 222000, Jiangsu, China

**Keywords:** *Bdellovibrio* powder, characterization, *Vibrio*, *Penaeus vannamei*

## Abstract

Liquid preparations of bdellovibrios are currently commercialized as water quality improvers to control bacterial pathogens in whiteleg shrimp *Penaeus vannamei*. However, the efficacy of these liquid preparations is significantly impaired due to a dramatic loss of viable cells during long-term room temperature storage. Thus, new formulations of bdellovibrios are greatly needed for high-stablility room-temperature storage. In the present study, the encapsulated powder of *Bdellovibrio* sp. strain F16 was prepared using spray drying with 20 g L^−1^ gelatin as the coating material under a spray flow of 750 L h^−1^, a feed rate of 12 mL min^−1^, and an air inlet temperature of 140 °C. It was found to have a cell density of 5.4 × 10^7^ PFU g^−1^ and to have spherical microparticles with a wrinkled surface and a diameter of 3 μm to 12 μm. In addition, the encapsulated *Bdellovibrio* powder presented good storage stability with its cell density still remaining at 3.5 × 10^7^ PFU g^−1^ after 120 days of room-temperature storage; it was safe for freshwater-farmed whiteleg shrimp with an LD_50_ over 1200 mg L^−1^, and it exhibited significant antibacterial and protective effects at 0.8 mg L^−1^ against shrimp-pathogenic vibrios. To our knowledge, this is the first report on a promising *Bdellovibrio* powder against shrimp vibrios with high stable room-temperature storage.

## 1. Introduction

The whiteleg shrimp *Penaeus vannamei* is one of the most important farming shrimp species around the world and is extensively cultivated in Central and South America, USA, East and South East Asia, Middle East, and Africa [1]. However, vibriosis has become a major challenge in shrimp aquaculture because of the lack of effective and safe control agents [2]. For example, infections caused by *Vibrio parahaemolyticus*, *Vibrio cholerae*, and *Vibrio vulnificus* have resulted in significant economic losses in shrimp farming regions [3,4,5]. Thus, new agents to control vibriosis are needed for the sustainable development of the shrimp farming industry.

Predatory bdellovibrios are strongly considered to be an alternative source of antibiotics [6] and have been reported to have the potential to control shrimp pathogens such as *Vibrio cholerae* [4] and *Vibrio parahaemolyticus* [7]. Currently, the liquid preparations of bdellovibrios that are used in shrimp aquaculture are commercialized as water quality improvers and are widely available on the market [8]. However, the efficacy of these commercial liquid preparations has been significantly impaired as a result of the massive loss of viable cells during long-term room temperature storage [9]. Hence, effective strategies should be adopted to enhance the stability of viable cells during storage at room temperature.

Spray drying is a most promising encapsulation technique to prolong bacterial survivals in foods or under stress conditions [10,11]. Nowadays, probiotics such as *Lactobacillus acidophilus*, *Lactobacillus rhamnosus*, and *Bifidobacterium adolescentis* have been encapsulated by spray drying to improve the viability of probiotic cells during storage [12,13,14]. The encapsulated bacteriophage powder prepared by spray drying has also been commercialized [15]. However, the encapsulation of bdellovibrios by spray drying has never been reported. In this study, we optimized the production of the encapsulated *Bdellovibrio* powder by spray drying. The microparticles of the encapsulated *Bdellovibrio* powder were observed, and room-temperature storage stability, safety, as well as in vitro antibacterial and protective effects against freshwater-cultured whiteleg shrimp-pathogenic vibrios were further evaluated. To our knowledge, this is the first study to develop and characterize a promising *Bdellovibrio* powder with bactericidal activity against freshwater-cultured whiteleg shrimp-pathogenic vibrios.

## 2. Materials and Methods

### 2.1. Microorganisms and Reagents

*Bdellovibrio* sp. strain F16, previously isolated from the gut of Siberian sturgeon *Acipenser baerii* and identified by 16S rRNA gene sequencing (deposited in GenBank as accession number HQ225833) [16], was used in this study. *Escherichia coli* strain DH5α and three freshwater-cultured whiteleg shrimp-pathogenic vibrios (*Vibrio cholerae* strain BB31, *Vibrio parahaemolyticus* strain G1, and *Vibrio vulnificus* strain A2) were obtained from the National Pathogen Collection Center for Aquatic Animals, China. Filtered farm water used in our laboratory was produced by passing 5 m^3^ of water, which was obtained from Shanghai Yuye Shrimp Farming Co., Ltd., China, successively through 15-denier-size polyester fiber and polyurethane sponge in a water filtration device (Shanghai Haisheng Biotech. Co., Ltd.), with the flow rate of 0.24 m^3^ min^−1^ and the circulation rate of 8 times d^−1^ as recommended by Luo et al. (2008). [17]. Reagents were of analytical grade from the Sinopharm Chemical Reagent Co., Ltd., China.

### 2.2. Preparation of Bdellovibrio Cells

Prior to the preparation of *Bdellovibrio* sp. strain F16 cells, *E. coli* strain DH5α was inoculated into 100 mL of nutrient broth and cultivated under the conditions described by Yu et al. (2010) [18]. *Bdellovibrio* sp. strain F16 was inoculated into 100 mL of diluted nutrient broth (DNB) [19] containing the prey *E. coli*, incubated at 30 °C with shaking at 180 rpm for 72 h [20]. The culture filtrate was prepared by a double process of 0.22-μm-pore-size membrane filtration and was carefully examined by transmission electron microscopy [21] and bacteriophage plaque assay [22] to determine that no bacteriophage was present. The cells of *Bdellovibrio* sp. strain F16 were obtained as described by Cao et al. (2012) [16]. The enumeration of *Bdellovibrio* sp. strain F16 cells was conducted using the double-layer agar plating method [23] and was recorded as plaque-forming units (PFU) per milliliter.

### 2.3. Optimization of the Spray Drying Process for Bdellovibrio Cells

Optimization of the spray-drying process for *Bdellovibrio* sp. strain F16 cells was performed using a single-factor experiment and an orthogonal test as recommended by Wang et al. (2006) [24]. Prior to the spray drying, gelatin was chosen as the coating polymer according to Guergoletto et al. (2017) [25] and was dissolved in distilled water, sterilized at 121 °C for 15 min and maintained at 50 °C according to Gu et al. (2015) [26]. *Bdellovibrio* sp. strain F16 was assayed for its thermal stability in a water bath at 50 °C for 60 min according to Gao et al. (2016) [27], and enumeration of *Bdellovibrio* sp. strain F16 was conducted at regular intervals of twenty minutes using the double-layer agar plating method [23] after serial ten-fold dilution [28]. Gelatin concentration, spray flow, feed rate, and air inlet temperature that significantly affect the process for spray drying of probiotic cells [10,29] were selected for further analysis. In the single-factor experiment, three replicates of 200 mL of *Bdellovibrio* sp. strain F16 (5.0 × 10^6^ PFU mL^−1^) were incorporated into 800 mL of the coating materials (10, 20, 30, 40, 50 g L^−1^ of gelatin) at 50 °C and mixed continuously with a magnet mixer at a speed of 100 rpm [29], then the mixtures were, respectively, fed into a laboratory spray dryer (Model SY-6000, Shanghai Shiyuan Bio-engineering Equipment Co. Ltd., China) under conditions of spray flows (650, 700, 750, 800, and 850 L h^−1^), feed rates (6, 8, 10, 12, and 14 mL min^−1^), and air inlet temperatures (120, 130, 140, 150, and 160 °C). In the orthogonal test, based on the single-factor analysis, a L_9_(3^4^) orthogonal design was performed in triplicate to further optimize the spray-drying process. In order to escape from contamination of the spray-dried powder, the compressed air supplied to the spray dryer was filtered using a 0.2 µm-pore-size membrane filter (Source Filter Technology (Hangzhou) Co., Ltd., China) to remove bacteria. The spray-dried samples were collected and stored at a room temperature of 25 °C [30]. The enumeration of *Bdellovibrio* sp. strain F16 cells in the spray-dried powder was performed using the double-layer agar plating method [23] after serial ten-fold dilution [28] and recorded as PFU per gram.

### 2.4. Microparticle Observation of Bdellovibrio Powder

The microparticles of *Bdellovibrio* powder were examined by scanning electron microscopy (S-3400, Hitachi, Japan), as recommended by Ann et al. (2007) [31]. Dry powder was fixed on metal stubs with double-sided tape and coated with gold in a high-vacuum evaporator. Images were taken at a reduced pressure of 9.75 × 10^−5^ Torr and at an accelerating voltage of 10 kV [29]. In addition, *Bdellovibrio* sp. strain F16 released from the microparticles was examined by transmission electron microscopy. Briefly, *Bdellovibrio* powder was dispersed in autoclaved deionized water as described by Song et al. (2014) [32], then the mixture was dripped onto a copper net, negatively stained with 0.5% sodium phosphotungstic acid (pH 7.0) for 1 min as recommended by Falk et al. (1997) [33] and observed under transmission electron microscope (HT7800, Hitachi, Japan). Its plaque forming ability was also examined using the double-layer agar plating method [23], as recommended by Wen et al. (2009) [34].

### 2.5. Storage Stability of Bdellovibrio PowderAssay

The storage stability of *Bdellovibrio* powder was performed in triplicate and was checked by detecting the survival of *Bdellovibrio* sp. strain F16 in the spray-dried powder stored at a room temperature of 25 °C as recommended by Li et al. (2009) [35]. During storage for one hundred and twenty days, the enumeration of *Bdellovibrio* sp. strain F16 in the spray-dried powder stored at room temperature was conducted at regular intervals of fifteen days using the double-layer agar plating method [23] after serial ten-fold dilution [28]. Another commercial liquid preparation of *Bdellovibrio* sp. strain F16 (5.4 × 10^7^ PFU mL^−1^), which was obtained from Shanghai Bio-Green Biotechnology Co. Ltd., China, was stored under the same conditions and served as the control.

### 2.6. The in Vitro Antibacterial Effect of Bdellovibrio Powder against Shrimp Pathogenic Vibrios Assay

Prior to this assay, suspensions of *V. cholerae* strain BB31, *V. parahaemolyticus* strain G1, and *V. vulnificus* strain A2 were, respectively, prepared as described by Lin et al. (2007) [36]. The gamma-irradiation-killed *Bdellovibrio* sp. strain F16 was obtained according to Altay et al. (2018) [37], and its powder was prepared using the spray-drying process optimized above. The antibacterial effect of *Bdellovibrio* powder against the whiteleg shrimp-pathogenic vibrios was conducted in triplicate and carried out in glass flasks. In the treatment flasks, the *Bdellovibrio* powder and a suspension of a pathogenic strain were independently inoculated into 200 mL of autoclaved filtered farm water to final concentrations of 0.4, 0.8 mg L^−1^, and 5.0 × 10^6^ CFU mL^−1^. The mixtures were then incubated at 30 °C with shaking at 180 rpm for 5 days. In the positive control flasks, the gamma-irradiation-killed *Bdellovibrio* powder (with a final concentration of 0.8 mg L^−1^) and a suspension of a pathogenic strain (with a final concentration of 5.0 × 10^6^ CFU mL^−1^) were independently inoculated into autoclaved filtered farm water and incubated as mentioned above. In the negative control flasks, only a pathogenic strain was inoculated in autoclaved filtered farm water and incubated as mentioned above. The cell density of the pathogenic strains was measured at intervals of one day by spread-plate counts on thiosulfate-citrate-bile salts-sucrose (TCBS) agar [38].

### 2.7. Safety of Bdellovibrio Powder Assay

The safety assay of *Bdellovibrio* powder was performed according to the Ministry of Agriculture of China (2003) [39] and consisted of one control and six treatment groups. Two hundred and ten healthy freshwater-cultured whiteleg shrimp (averaging 0.55 ± 0.13 g in weight) were obtained from a shrimp farm in Lianyungang, Jiangsu province, China and maintained in a twenty-one glass aquaria (76 cm × 50 cm × 48 cm) supplied with the same aerated filtered farm water with an initial pH of 7.90, 6.5 mg L^−1^ of dissolved oxygen, 0.12 mg L^−1^ of total ammonia, and 0.01 mg L^−1^ of nitrite at 28 °C throughout the experiment. Their health status was assessed through a careful examination of external appearance, gut condition, growth situation, physical behavior, and feeding trends, as recommended by the Marine Products Export Development Authority and the Network of Aquaculture Centers in Asia-Pacific (2003) [40]. Each aquarium, containing 100 L of the same farm water without water recirculation, was stocked with 10 healthy shrimp selected at random. In the treatment groups (three aquaria per group), *Bdellovibrio* powder from the same batch was independently added into the aerated filtered farm water to final concentrations of 200, 400, 600, 800, 1000, and 1200 mg L^−1^. Another three aquaria of healthy shrimp, which were exposed to the same experimental conditions, served as the control. The mortality and disease signs were observed and recorded every day for four days in the test shrimp without feeding and water change [29]. The mean lethal dose (LD_50_) value was calculated according to the graphical probit method, as recommended by Ogbuagu and Iwuchukwu (2014) [41].

### 2.8. Protective Effect of Bdellovibrio Powder Assay

The protective effect assay of *Bdellovibrio* powder consisted of three control and three treatment groups. One hundred and eighty healthy freshwater-cultured whiteleg shrimp (averaging 0.62 ± 0.11 g in weight) were obtained from Lianyungang, Jiangsu province, China, and maintained in eighteen glass aquaria (76 cm × 50 cm × 48 cm) supplied with the same aerated filtered farm water with an initial pH of 7.64, 6.6 mg L^−1^ of dissolved oxygen, 0.10 mg L^−1^ of total ammonia, and 0.01 mg L^−1^ of nitrite at 28 °C throughout the experiment. Their health status was assessed through a careful examination, as described above. Each aquarium, containing 100 L of the same farm water without water recirculation, was stocked with 10 healthy shrimp selected at random. In the treatment groups (three aquaria per group), *Bdellovibrio* powder from the same batch was directly added into the 100 L of aerated filtered farm water to a final concentration of 0.8 mg L^−1^, as determined above. Immediately thereafter, all of the shrimp were challenged by immersion through continuous exposure to the same batch of freshly cultured shrimp-pathogenic strain (*V. cholerae* strain BB31, *V. parahaemolyticus* strain G1, *V. vulnificus* strain A2) at a final concentration of 5 × 10^6^ CFU mL^−1^, as recommended by Zhang et al. (2009) [42], Saulnier et al. (2000) [43], and Cao et al. (2015) [4]. In the control groups (three aquaria per group), the shrimp were exposed to the same experimental conditions and only challenged by immersion with the pathogenic strain at the final cell density above. The mortality was observed and recorded every day for six days in the test shrimp without feeding and water change [29]. Any dead shrimp were immediately removed and sampled to re-isolate and confirm specific mortality by the challenge strain. Relative percentage survivals were calculated according to Baulny et al. (1996) [44].

### 2.9. Statistical Analysis

Statistical analysis was carried out using the statistical software SPSS 15.0 (SPSS, Inc.) to observe the difference in each assay. All of the data were presented as the mean ± standard deviation (SD) for the indicated number of each assay. Differences were considered statistically significant at *p* < 0.05 using analysis of variance according to Duncan’s test.

### 2.10. Ethics Statements

The experimental protocol strictly followed the guidelines for ethical review of animal welfare and the general requirements for animal experiment, China. The present experiment was approved by the Institutional Animal Ethics Committee (approval date: 18 Feb. 2016) of Shanghai Ocean University with the permission No. 20171025 dated on 7 Feb. 2017.

## 3. Results

### 3.1. Optimization of the Spray-Drying Process for Bdellovibrio Cells

*Bdellovibrio* sp. strain F16 possessed good thermal stability, as shown in Appendix A, indicating that *Bdellovibrio* sp. strain F16 could survive at 50 °C. In addition, *Bdellovibrio* sp. strain F16 cells could be well encapsulated by spray drying with 20 g L^−1^ of gelatin under spray flows of 650 to 750 L h^−1^, feed rates of 8–12 mL min^−1^, and an air inlet temperature of 140 °C (Figure 1). On the basis of the data, the orthogonal test was further designed to optimize the spray-drying process for the *Bdellovibrio* sp. strain F16 cells. The result showed that the cell densities of *Bdellovibrio* sp. strain F16 were detected to be 1.45 × 10^6^ to 3.02 × 10^7^ PFU g^−1^ in the spray-dried powder prepared under different spray-drying conditions (Table 1). The ranking of the four factors in the orthogonal tests was B (spray flow) > A (gelatin concentration) > C (feed rate) > D (air inlet temperature), and the individual levels within each factor were ranked as: A: 2 > 3 > 1; B: 3 > 2 > 1; C: 3 > 1 > 2; D: 2 > 3 > 1. The optimal combination for the spray drying of *Bdellovibrio* sp. strain F16 was A_2_B_3_C_3_D_2_ according to the analysis of orthogonal design assistant software version 3.1 (Analytical Software, Internet), indicating that *Bdellovibrio* powder could be spray dried most effectively with 20 g L^−1^ gelatin as the coating polymer under the spray flow of 750 L h^−1^, feed rate of 12 mL min^−1^, and air inlet temperature of 140 °C. The *Bdellovibrio* powder prepared under the optimal conditions above was finally demonstrated to have the highest cell density of 5.4 × 10^7^ PFU g^−1^ and was further investigated in this study.

### 3.2. Microparticle Morphology of Bdellovibrio Powder

The microparticle morphology of *Bdellovibrio* powder is shown in Figure 2. The microparticles presented spherical shapes with a wrinkled surface and a diameter of 3 to 12 μm, which is a characteristic feature of spray-dried microparticles containing probiotic cells [25]. Free cells were not visualized under scanning electron microscopy, indicating that all of the *Bdellovibrio* sp. strain F16 cells were entrapped inside the microparticles. Additionally, the vibroid-shaped *Bdellovibrio* sp. strain F16 released from the microparticle was observed under transmission electron microscope after the addition of water (Figure 3), which showed the typical morphological features of bdellovibrios [45]. Besides, it could form round plaques as described by Williams et al. (1980) [46] on the double-layer agar plate after incubation for 48 h (Appendix A), which differed from bacteriophage plaques that developed within 24 h [47]. These findings further indicated the presence of bdellovibrios and the absence of bacteriophages in the microparticles.

### 3.3. Storage Stability of Bdellovibrio Powder

The survival of *Bdellovibrio* sp. strain F16 in the spray-dried powder and liquid preparation during room temperature storage is shown in Figure 4. The result showed that the survival of *Bdellovibrio* sp. strain F16 was greatly improved in the encapsulated powder, compared with that in the liquid preparation, with 3.5 × 10^7^ PFU g^−1^ still alive after 120 days of room-temperature storage. However, a dramatic decline in the cell density of *Bdellovibrio* sp. strain F16 was observed in the liquid preparation, showing only 4.4 × 10^3^ PFU mL^−1^ of *Bdellovibrio* sp. strain F16 alive after 120 days of room-temperature storage. This indicates that the encapsulated *Bdellovibrio* powder presents better storage stability than the liquid preparation.

### 3.4. The in Vitro Antibacterial Effect of Bdellovibrio Powder against Shrimp-Pathogenic Vibrios

The in vitro antibacterial effect of *Bdellovibrio* powder against shrimp-pathogenic vibrios is shown in Figure 5. The result demonstrated that *Bdellovibrio* powder at 0.8 mg L^−1^ clearly inhibited the growth of the shrimp-pathogenic vibrios better than *Bdellovibrio* powder at 0.4 mg L^−1^ did (*p* < 0.05). The growth of the pathogenic *V. cholerae*, *V. parahaemolyticus*, and *V. vulnificus* treated with 0.4 and 0.8 mg L^−1^ of *Bdellovibrio* powder was significantly inhibited, respectively, showing a reduction of 53.28% (*p* < 0.05) and 98.80% (*p* < 0.05), 95.32% (*p* < 0.05) and 99.97% (*p* < 0.05), 98.62% (*p* < 0.05) and 99.91% (*p* < 0.05) in the cell density after treatment for five days as compared with the negative control. However, a slight increase was observed in the cell densities of the pathogenic *V. cholerae*, *V. parahaemolyticus*, and *V. vulnificus* treated with the gamma-irradiation-killed *Bdellovibrio* powder after treatment for five days as compared with the negative control. In addition, *Bdellovibrio* powder at 0.8 mg L^−1^ had a stronger antibacterial effect against the pathogenic vibrios, and the cell densities of the pathogenic *V. cholerae*, *V. parahaemolyticus*, and *V. vulnificus* were significantly reduced by 58.2% (*p* < 0.05), 97.6% (*p* < 0.05), and 89.8% (*p* < 0.05) after five days of treatment as compared with the initial cell densities. Thus, *Bdellovibrio* powder was recommended for use at 0.8 mg L^−1^ to control the pathogenic vibrios.

### 3.5. Safety of Bdellovibrio Powder

No acute mortality or any visible disease signs were observed in the test whiteleg shrimp treated with 200 to 1200 mg L^−1^ of *Bdellovibrio* powder (data not shown). It is concluded that the LD_50_ value of *Bdellovibrio* powder is estimated to exceed 1200 mg L^−1^.

### 3.6. Protective Effect of Bdellovibrio Powder

The protective effect of *Bdellovibrio* powder against the challenge of vibriosis in shrimp is shown in Figure 6. The result showed that *Bdellovibrio* powder at 0.8 mg L^−1^ could confer significant protection against *Vibrio* infections in freshwater-farmed *P. vannamei*. The cumulative mortality of shrimp treated with 0.8 mg L^−1^ of *Bdellovibrio* powder was 43.3% (*p* < 0.05), 70.0% (*p* < 0.05), and 53.4% (*p* < 0.05) lower than that in the control after the challenge with the pathogenic *V. cholerae*, *V. parahaemolyticus,* and *V. vulnificus*. The relative percentage survivals of 61.9%, 80.8%, and 69.6% were obtained against the challenge with the *V. cholerae*, *V. parahaemolyticus*, and *V. vulnificus* strains in shrimp for six days. The death of all of the test shrimp was caused by the challenge strains, as determined by bacterial isolation and identification (data not shown).

## 4. Discussion

Currently, liquid preparations of bdellovibrios have been successfully developed and widely applied in aquaculture [48,49]. However, the spray-dried *Bdellovibrio* powder has been seldom documented. In the present study, we are the first to report a promising spray-dried *Bdellovibrio* powder with bactericidal activity against shrimp-pathogenic vibrios.

Various parameters are reported to potentially affect the process for spray drying of probiotic cells [29]. Hence, the optimization of the spray-drying process is critical for the preparation of encapsulated probiotic powder. During spray drying, the coating material, spray flow, feed rate, and air inlet temperature are considered to be the most important factors that influence the encapsulation of probiotic cells [10,29]. Gelatin is known as a safe coating material with no cytotoxicity [50], which has good ability to reduce the heat transfer to the viable cells during spray drying [51]. Therefore, in the present study, gelatin was employed as the coating material, as recommended by Guergoletto et al. (2017) [25]. Furthermore, gelatin concentration, spray flow, feed rate, and air inlet temperature were selected to optimize the spray-drying process by orthogonal design on the basis of the single-factor test, as recommended by Wang et al. (2006) [24]. Using the encapsulated cell density as a key indicator for evaluation of the quality of encapsulated powders [52], the optimized spray-drying process for bdellovibrios was acceptable, as indicated by the high density of encapsulated cells. This makes spray drying an alternative technique for the preparation of encapsulated *Bdellovibrio* powder.

The spray-dried encapsulated microparticles containing *Bifidobacterium* and *Lactobacillus reuteri* cells have been documented to possess characteristic morphological features, i.e., spherical with a wrinkled surface [10,25]. In our study, the encapsulated *Bdellovibrio* microparticles were also found to be spherical with a wrinkled surface, in accordance with that observed by O’Riordan et al. (2001) [10] and Guergoletto et al. (2017) [25]. This is probably attributed to the inherent characteristics of coating polymers [10], as well as the high temperature used in the drying chamber and the atomized droplets size [53]. In addition, encapsulation of probiotics by spray drying has been confirmed as an effective approach for prolonging cell stability during storage [10]. In our study, a better survival of *Bdellovibrio* sp. strain F16 was also found in the encapsulated powder than the liquid preparation during room temperature storage. This may be due to the fact that encapsulation can significantly reduce the environmental stress-induced cell death [54].

In order to make its application safe, the candidate probiotic product has to be evaluated for its safety [8,55]. Many studies have reported that bdellovibrios are considered as good probiotics for food and environmental safety [8], which have no cytotoxicity to fish cells [56]; exhibit no hemolytic activity [16]; show no virulence for fish, shrimp, and mice [16,42,57]; and reduce ammonia, nitrite, sulphide, and population of bacterial pathogens in aquaculture water [58,59]. In particular, bdellovibrios are present and abundant only in healthy humans [60], which are positively correlated with gut microbiome diversity [61]. Food with bdellovibrios can act as drivers of gut microbial diversity with no pathogenicity and toxicity to humans [61,62,63], which can restore gut microbiomes and prevent dysbiosis to improve human health [61]. In the present study, the LD_50_ value of the encapsulated *Bdellovibrio* powder to whiteleg shrimp exceeded 1200 mg L^−1^. According to the toxicity rating criteria, as suggested by Yoshimura and Endoh (2005) [64], the encapsulated *Bdellovibrio* powder can be categorized into a practically nontoxic class (LD_50_ > 100 mg L^−1^), indicating that the encapsulated *Bdellovibrio* powder is a potential safe candidate for use in shrimp aquaculture.

In addition, to consider the encapsulated *Bdellovibrio* powder as a biodisinfectant against shrimp-pathogenic vibrios, it is essential to obtain data on its antibacterial and protective effects against shrimp-pathogenic vibrios. In our study, the encapsulated *Bdellovibrio* powder at 0.8 mg L^−1^ was found to significantly reduce the cell density of the pathogenic *V. cholerae*, *V. parahaemolyticus*, and *V. vulnificus* by 58.2% (*p* < 0.05), 97.6% (*p* < 0.05), and 89.8% (*p* < 0.05) after five days of treatment as compared with the initial cell density. However, the cell density of the pathogenic vibrios did not constantly decline when treated with 0.8 mg L^−1^ of the *Bdellovibrio* powder. In view of the fact that gelatin could contribute to the bacterial growth [65], the reduction of the shrimp-pathogenic vibrios is presumably due to live bdellovibrios and gamma-irradiation-susceptible predatory enzymes or other biomolecules. In addition, relative percentage survivals of 61.9%, 80.8%, and 69.6% were also obtained at 0.8 mg L^−1^ of the encapsulated *Bdellovibrio* powder against the challenge with the pathogenic *V. cholerae*, *V. parahaemolyticus*, and *V. vulnificus* in shrimp for six days. This is probably due to a primitive immune response in the shrimp induced by bdellovibrios and other immunogenic substances from the powder, which could immediately prime the shrimp to resist bacterial infections. According to the criteria for assessing the effect of probiotic preparations used in aquaculture [39], the encapsulated *Bdellovibrio* powder can be considered a significant effective biodisinfectant against the shrimp-pathogenic vibrios.

In conclusion, the spray-drying process was optimized in our study through single-factor experiment and orthogonal test to prepare the encapsulated *Bdellovibrio* powder. The safety, significant antibacterial, and protective effects towards whiteleg shrimp-pathogenic vibrios demonstrated that the encapsulated *Bdellovibrio* powder could be used as a potential biodisinfectant in freshwater-farmed whiteleg shrimp.

## Figures and Tables

**Figure 1 microorganisms-07-00244-f001:**
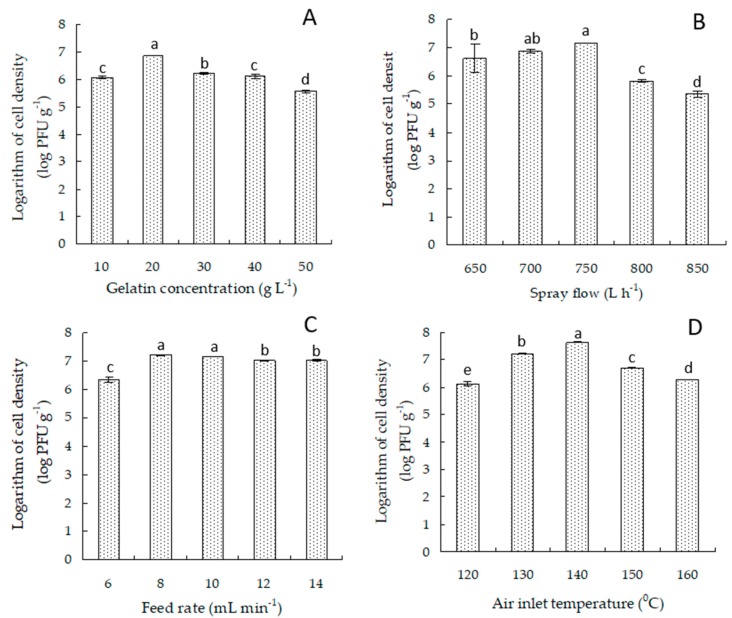
Effects of gelatin concentration (**A**), spray flow (**B**), feed rate (**C**), and air inlet temperature (**D**) on cell density of *Bdellovibrio* sp. strain F16 in the spray-dried powder. Data presented as the mean of triplicate spray-drying trials and standard deviations (SD) are indicated by the vertical bars. Bars with different lowercase letters are statistically different (*p* < 0.05).

**Figure 2 microorganisms-07-00244-f002:**
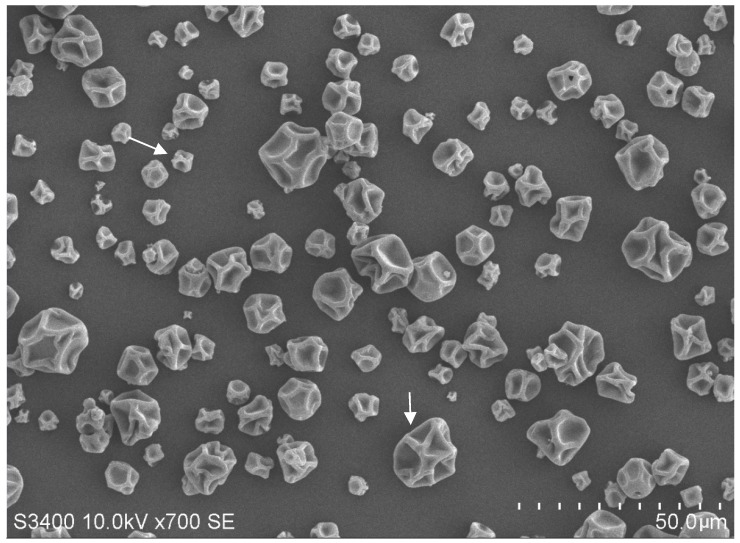
Microparticles of *Bdellovibrio* powder. Arrows show the spherical microparticles with a wrinkled surface.

**Figure 3 microorganisms-07-00244-f003:**
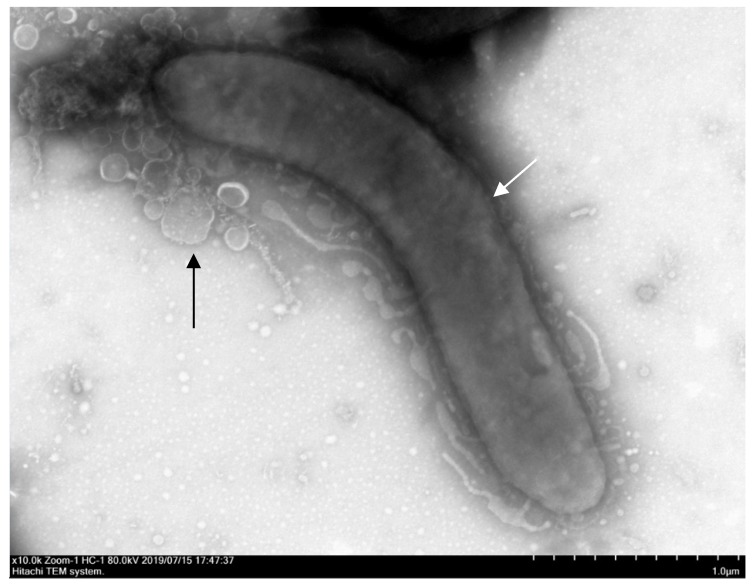
Transmission electron microscopy image of *Bdellovibrio* sp. strain F16 released from the microparticles after the addition of water. White arrow shows the vibroid-shaped *Bdellovibrio* cell, black arrow shows the amorphous gelatin gel particles around the cell.

**Figure 4 microorganisms-07-00244-f004:**
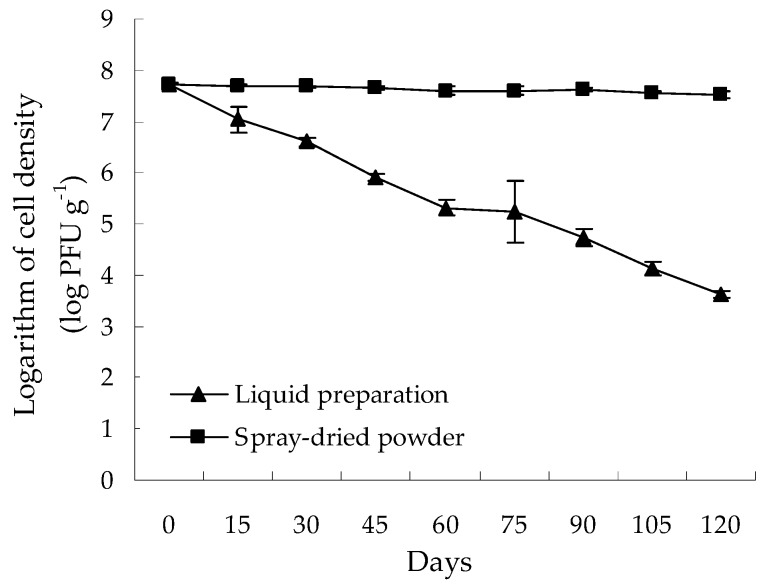
Survival of *Bdellovibrio* sp. strain F16 in the spray-dried powder and liquid preparation during room temperature storage. Data are presented as the mean ± SD.

**Figure 5 microorganisms-07-00244-f005:**
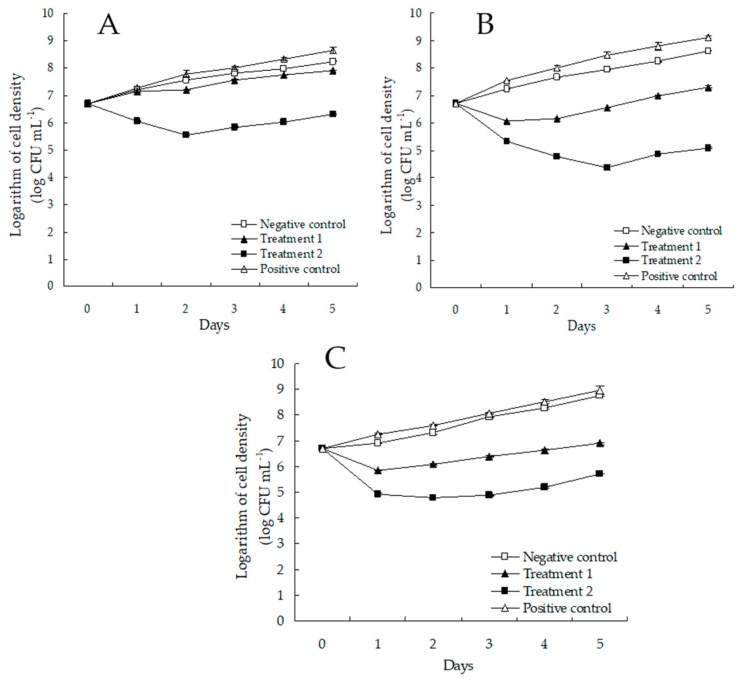
Antibacterial effect of *Bdellovibrio* powder against the shrimp-pathogenic *V. cholerae* (**A**), *V. parahaemolyticus* (**B**), *V. vulnificus* (**C**); negative control: 0 mg L^−1^
*Bdellovibrio* powder; treatment 1: 0.4 mg L^−1^
*Bdellovibrio* powder; treatment 2: 0.8 mg L^−1^
*Bdellovibrio* powder; positive control: 0.8 mg L^−1^ gamma-irradiation-killed *Bdellovibrio* powder. Data are presented as the mean ± SD. Any differences observed are considered statistically significant at *p* < 0.05 according to Duncan’s test.

**Figure 6 microorganisms-07-00244-f006:**
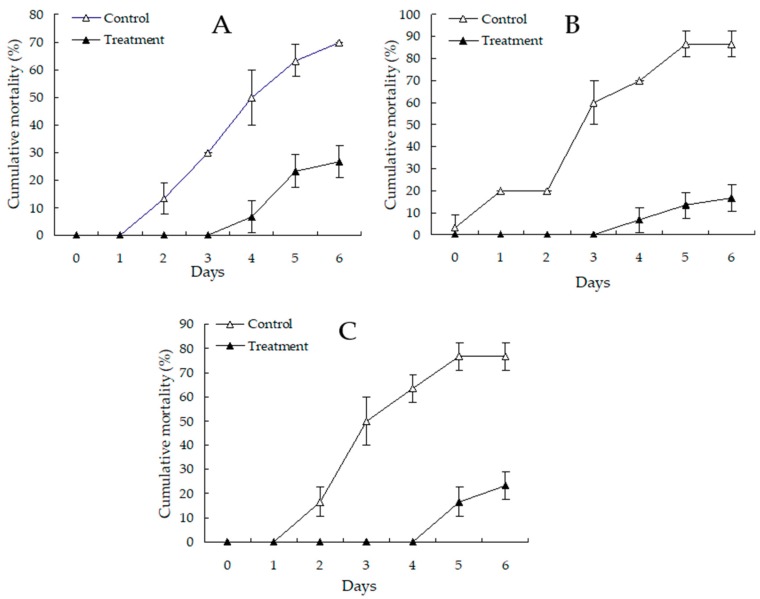
Protection of whiteleg shrimp by *Bdellovibrio* powder against the shrimp-pathogenic *V. cholerae* (**A**), *V. parahaemolyticus* (**B**), and *V. vulnificus* (**C**); control: 0 mg L^−1^
*Bdellovibrio* powder; treatment: 0.8 mg L^−1^
*Bdellovibrio* powder. Data are presented as the mean ± SD. Any differences observed are considered statistically significant at *p* < 0.05 according to Duncan’s test.

**Table 1 microorganisms-07-00244-t001:** L_9_ (3^4^) orthogonal design to investigate the effect of experimental factors on cell densities of *Bdellovibrio* sp. strain F16 in the spray-dried powder.

Test No.	A (g L^−1^)	B (L h^−1^)	C (mL min^−1^)	D (°C)	Cell Density (log PFU g^−1^)
1	15	650	8	135	6.16 ± 0.15 ^e^
2	15	700	10	140	6.91 ± 0.13 ^bcd^
3	15	750	12	145	7.12 ± 0.09 ^b^
4	20	650	10	145	6.74 ± 0.16 ^d^
5	20	700	12	135	7.48 ± 0.06 ^a^
6	20	750	8	140	7.63 ± 0.01 ^a^
7	25	650	12	140	6.77 ± 0.27 ^cd^
8	25	700	8	145	6.91 ± 0.02 ^bcd^
9	25	750	10	135	7.02 ± 0.11 ^bc^
*K*1	6.730	6.557	6.900	6.887	
*K*2	7.283	7.100	6.890	7.103	
*K*3	6.900	7.257	7.123	6.923	
*R*	0.553	0.700	0.233	0.216	

A, gelatin concentration; B, spray flow; C, feed rate; D, air inlet temperature. *K*1, *K*2 and *K*3 are the average scores of level 1, level 2 and level 3 for each factor. *R* is the range among the average scores for each factor. Data are presented as the mean ± deviations (SD). Values with different superscript letters in the column indicate statistically significant difference (*p* < 0.05).

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
