# Peer review of "Encapsulated Bdellovibrio Powder as a Potential Bio-Disinfectant against Whiteleg Shrimp-Pathogenic Vibrios"

_microorganisms, 2019, doi:10.3390/microorganisms7080244_

Round 1

Reviewer 1 Report

The formulation of spray dried powder containing live Bdellovibrio is potentially very important and of great interest to readers. So the topic is important.

However details in the manuscript are of concern and these need to be addressed to prove that the Bdellovibrio are actually live in the formulation. Without this the paper is not robust. This is essential for such a potentially important topic.

I am potentially excited by this discovery but we need the fuller evidence, with controls, to be presented to be sure. I respectfully request that this additional evidence is produced.

1) Evidence must be provided of live bdellovibrio emerging from the micro-particles, not simply the microparticle image in Figure 2.

Such evidence requires addition of liquid droplets to the microparticles (can be performed on the microscopy grids or on light microscope) and then microscopy pictures of the Bdellovibrio cells emerging live into those droplets.

2) It is surprising that the Bdellovibrio which are cultured at 30degrees C survive the 50 degrees C incubation with the spray drying agent (& the 140C air inlet temperatures). I acknowledge that the authors claim that Ref 21 indicates less heat transfer to the Bdellovibrio and they do survive the warm (50C) agar overlay for plaque assay plates but this is why evidence is required as in point 1).

3) It is possible that dead Bdellovibrio containing proteases or glycanases are present in the particles and that their enzymes alone have an effect on the pathogen. This would still give a diluteable factor as shown in Figure 4, but would not be due to live Bdellovibrio encapsulation.

Please can deliberately UV or heat killed Bdellovibrio be made into a power and used as a control in the enumeration assays and in the pathogen killing?

4) It is possible that a bacteriophage co-cultured, by chance, with the Bdellovibrio and is active against  the pathogen. 

This would still form plaques but they would be different to Bdellovibrio plaques. Please show side by side plaque plate photos from pure Bdellovibrio culture. 

Did you check on the electron microscope that no phage were present with the Bdellovibrio? Phage would survive the heated encapsulation process well?

I am sorry if these questions sound very direct.  Its just that the evidence is needed. I hope that the manuscript can be improved in this way and show directly that the live Bdellovibrio are present in the spray dried particles and active. I am excited to know about it.

Author Response

Dear reviewer, the changes have been made according to your suggestion. Please see the attachment. Many thanks.

Reviewer 2 Report

This is an interesting paper regarding the efficacy of encapsulated, spray-dried Bdellovibrio at reducing Vibrio in shrimp. The authors examined the safety of exposing Bdellovibrio to shrimp, which was important to include in the experimental design. This study provides a preliminary understanding and will serve as a good foundation of knowledge for future research.

General Comments

More information regarding statistical analysis should be included in the paper, and this will be specified below in the review.

Overall, this paper would benefit from a thorough edit for proper English and grammar; therefore, specific edits pertaining to English and grammar will not be identified in this review.

In general, more information is necessary regarding the well-being of the shrimp. They were observed daily, but what type of observations were noted? Was there data collected pertaining to these daily observations?

In each of the italicized section headings, Bdellovibrio should not be italicized. For example, Line 68 and every section heading thereafter.

The experimental design is not well-described, which makes it difficult to judge how sound this study is. For example, it doesn't seem that true replications of the experimental design were conducted, rather replicate sampling within a single replication seemingly took place. 

What is filtered farm water? Where was this water obtained? Was it prepared to simulate farm water? Purchased from a farm and, if so, how did the authors ensure homogeneity between tanks? 

What were the aquarium dimensions?

Perhaps the control shrimp, or a second set of control shrimp, should have been exposed to the gelatin powder without the Bdellovibrio. Do the authors think that the gelatin powder could have impacted shrimp and/or Vibrio survivability either positively or negatively? This should be addressed prior to publication.

Specific Comments

Lines 53-54: awkward, suggest revising.

Line 58: "...with the potential against..." suggest revising to be more specific. Do the authors mean to say predatory, bactericidal, etc.?

Line 91: more details should be provided about the filter. For example, what was the pore size?

Section 2.3: were statistical analyses conducted? 

Figure 1 and Table 1: statistical significance?

Line 132: what does "observed daily" mean? What observations were taken? What data was collected? What was the threshold for healthy vs. unhealthy? 

Lines 201-202: what statistical analyses were conducted to generate these p-values?

Figure 4: statistical significance between the treatments?

Lines 245-248: statement is a run-on sentence, which makes it confusing. Suggest revising.

Line 272: what statistical analyses generated these p-values?

Author Response

Dear reviewer,

The changes have been made according to your suggestion. Please see the attachment. Many thanks.

Round 2

Reviewer 1 Report

 The paper has been improved by adding gamma irradiated Bdellovibrio to powder particles in a control predation experiment. This shows that either live Bdellovibrio or gamma-irradiation-susceptible predatory enzymes or other biomolecules are responsible for the killing of the pathogens. There is a difference between the control and the two treatments but the two treatments are "used up" quite quickly. The predation experiments in vitro Fig4 show only a small reduction in Vibrio numbers which then flatlines around 105 these data can be interpreted as a non replicating antimicrobial chemical agent which is being used up- rather than replicating live predatory bacteria emerging from the particles. 

The paper still lacks direct proof of Bdellovibrio emerging from the particles. Despite the gamma irradiated control, no microscopic proof was presented that vibrio shaped flagellate Bdellovibrio bacteria emerge from the live powder particles when incubated with farmwater, and not from the gamma irradiated particles. Unfortunately the Figure S2 plaque plate image is unclear and does not at all substitute for this requirement. . This is a simple microscopic experiment for a light- phase contrast or electron microscope. It would be the simple proof that makes this an important paper and discovery. It would finish the paper beautifully.

Similarly the data in Figure 5 could be due to a primitive immune response by the shrimp to a large amount of dead but immunogenic cell wall and other materials from the Bdellovibrio in the particles. This immune response could immediately prime the shrimp to resist bacterial infections (by whole, not yet broken Vibrio)  by its innate immunity being induced by broken glycans of dead Bdellovibrio cell walls from the particles. Because the experimental days did not extend beyond 7 for Fig 5 it was not possible to see if the pathogen growth in the "Treatment" samples was delayed and eventually rose up to give 80% mortality say at 10 days or whether it never occurred.

I really hope this live bacterial proof can be addressed to complete what can be a very nice story.

There are two typos and one methods section that is unclear.

 Line 20 high-stability and not stable.

Line 111 Assay not Aassay

Line 74-75 does not explain what prey bacteria were used in the cross-streak method to test for bacteriophage. What was needed here was a double process of 0.22um filtration to hold back any Bdellovibrio (which are too large to pass) and to allow through phages and then test for any plaque clearing activity on the prey pathogens.

If the authors can show that live Bdellovibrio are present in and emerge from the particles that would be good scientifically. Then they need to provide a food and environmental safety consideration into their discussion as potentially they would be producing a food organism - shrimp- that is now deliberately treated with live bacteria and also releasing large numbers of live bacteria (albeit an environmental isolate), into open fish ponds where humans and animals are exposed to them. 

This journal asks reviewers to consider if there are ethical issues to addess- yes application of live Bdellovibrio to fish farms in particles would be an ethical issue at a time of spread of antibiotic resistance- from farmed animals with conventional antibiotics to humans. This is because Bdellovibrio may be used to treat human infections too but some kinds of Bdellovibrio resistances could be selected for, by human contact with treated ponds or aerosolised water from those ponds, if they are released at large numbers for fish farming.  The authors are claiming live Bdellovibrio release here (hopefully supported by the microscopy asked for). It is prudent and proper, having done so, to include a statement about a need for careful and regulated use to satisfy ethical use criteria.

Author Response

Dear prof.,

Attached is the revision in the revised manuscript according to your opinion.

Reviewer 2 Report

This revised manuscript is much-improved from the original version and the authors should be commended for such a thorough, prompt revision.

Some issues still remain that must be addressed prior to publication:

The authors need to do a better job of describing their replications. For example, in section 2.7, they state the work was completed in triplicate. So does that mean three replications? For each replicate, were there 21 treatment aquaria and 3 control aquaria, and then this was repeated three times total, for a total of 72 aquaria overall? The design would be best if three different "lots" of the Bdellovibrio powder were used, with 1 unique lot for each replication. These are the types of details that are unclear and must be clarified. Please clarify accordingly for all sections where replicates are described. Another example is section 2.3.

Regarding the filtered farm water, additional details are still needed. What makes your lab-prepared water "farm water" rather than just "water filtered by a filtration device?" How would your reader replicate what you did in their own experiment? Enough details are not provided for someone to duplicate your steps.

Including the gamma-irradiation-killed Bdellovibrio as a positive control improves the experimental design and adding this information to the manuscript is very helpful. One thing that is not clear is why the authors state that this control "had a stimulated effect" on the Vibrio growth. While this control had the highest population, it was very similar in population to the negative control. Furthermore, can this statement be substantiated with statistical analyses? My guess is that the positive and negative control populations would be statistically the same, so would the statement that the gelatin powder "stimulated" the Vibrio growth be correct then? Or, what were the authors comparing to when making this statement?

Author Response

Dear prof.,

Many thanks for your kind advice. Attached is the revision in the revised manuscript according to your opinions.

Round 3

Reviewer 1 Report

The image of the bacterium does not show any characteristics of a Bdellovibrio. It is not vibroid. Its does not have the characteristic wave amplitude diminishing to the tip flagellar shape. The particles are no zok.ed outside on the other paper image. This provides no evidence that Bdellovibrio are in the particles. Please keep trying to show this. This could be anything like a Pseudomonas a Rhodobacter but not a Bdellovibrio. I'm sorry but the proof is important. Use lower concentration of negative stain collaborate with others to get images. O.5% sodium phosphotungstate pH 7 for example.

 Also no safety statement of the right kind was added although a different safety statement was added. The point t is that eating food with Bdellovibrio in it will change the microbiome of the human gut and the microbiome of the human gut is crucial to many health and wellbeing features..there are many new papers on the microbiome and obesity and mental health..this kind of statement is needed

Reviewer 2 Report

More details are needed about preparing farm water. Is there a citation that could be included to support these methods? What type of water did you start with? How much water? These details are still unclear and the study is not reproducible.

The authors have partially addressed concerns regarding triplicate vs. replicate. However, one point of clarification still remains: when using the word replicate, what exactly was replicated? It is clear that the aquaria were replicated, but for each replication was a fresh batch of Vibrio grown? Was a fresh batch of farm water prepared? Were the treatment powders from the same batch/or different? If all 21 aquaria were observed/tested at the same time, but each set of 7 treatment aquaria represented a unique replication, then replication is the correct word. If one batch of farm water and Vibrio cultures were prepared for all 21 aquaria, then that would not be a true replication and would, instead, represent triplicate samplings. Please clarify.

Regarding the changes in wording for the differences between positive and negative controls in Figure 4, it is important for the authors to address whether or not these significant differences are actually meaningful. It is great that the statement is backed up with statistical analyses; however, it is also important to address whether or not these differences are meaningful from a biological sense. In general, if a difference in population is within the same log or 1/2 log, is this actually meaningful? Or, do you just have tight data and/or a lot of data points that decrease standard error, resulting in statistical differences. What do the authors think about this in terms of their data? 

Round 4

Reviewer 2 Report

This version has addressed my concerns.

Author Response

Thank you very much for your insightful comments on the manuscript.